# Influence of Solvent Polarity on the Conformer Ratio of Bicalutamide in Saturated Solutions: Insights from NOESY NMR Analysis and Quantum-Chemical Calculations

**DOI:** 10.3390/ijms25158254

**Published:** 2024-07-28

**Authors:** Valentina V. Sobornova, Konstantin V. Belov, Michael A. Krestyaninov, Ilya A. Khodov

**Affiliations:** G.A. Krestov Institute of Solution Chemistry, Russian Academy of Sciences, Ivanovo 153045, Russia

**Keywords:** spatial structure, bicalutamide, conformer populations, NMR, NOESY

## Abstract

The study presents a thorough and detailed analysis of bicalutamide’s structural and conformational properties. Quantum chemical calculations were employed to explore the conformational properties of the molecule, identifying significant energy differences between conformers. Analysis revealed that hydrogen bonds stabilise the conformers, with notable variations in torsion angles. Conformers were classified into ‘closed’ and ‘open’ types based on the relative orientation of the cyclic fragments. NOE spectroscopy in different solvents (CDCl_3_ and DMSO-d_6_) was used to study the conformational preferences of the molecule. NOESY experiments provided the predominance of ‘closed’ conformers in non-polar solvents and a significant presence of ‘open’ conformers in polar solvents. The proportions of open conformers were 22.7 ± 3.7% in CDCl_3_ and 59.8 ± 6.2% in DMSO-d_6_, while closed conformers accounted for 77.3 ± 3.7% and 40.2 ± 6.2%, respectively. This comprehensive study underscores the solvent environment’s impact on its structural behaviour. The findings significantly contribute to a deeper understanding of conformational dynamics, stimulating further exploration in drug development.

## 1. Introduction

Multicomponent crystals, with their potential to develop new forms of chemical compounds with biological activity, are a fascinating area of research. The challenge of enhancing the solubility and bioavailability of these compounds has led to the promising task of optimising methods for modifying known solid forms and creating new ones, such as salts, co-crystals, crystal solvates, and polymorphs [1,2,3,4,5,6]. According to the Biopharmaceutical Classification System (BCS), which exhibits low water solubility and high membrane permeability, compounds of class II are particularly intriguing for modification [7,8,9,10,11]. Critical factors in the formation processes of multicomponent crystals and polymorphs include the preparation environment (solvent) characteristics, state parameters, and their combined influence on nucleation processes [12]. However, as noted in [13], the basic principles for selecting solvents to obtain different solid forms of pharmaceutical compounds still need to be solved.

Researchers [13,14,15,16,17] have employed a trial-and-error method with solvents and their combinations to adjust solvation pathways to achieve the desired forms of various pharmaceutical compounds. Another approach [14,15,16,17] includes a combination of virtual and experimental screening methods, which has shown promise in the comprehensive search for methods of obtaining specific solid forms of pharmaceutical compounds [18,19]. Abramov et al. [20] and Hao’s group [21,22] researched how controlling conformer populations could aid in selecting solvents for screening polymorphic forms. Their work was reflected in previous studies on small molecules exhibiting polymorphism and solvatomorphism [23,24,25,26,27,28,29,30,31,32]. In our research, we use nuclear Overhauser effect spectroscopy (NOESY) to analyse the spatial structure of small molecules in saturated solutions [32,33,34,35,36,37,38,39]. This method is crucial for understanding the role of the solvent in forming specific crystalline forms. It has proven effective in determining the ratio of conformer groups in solutions [40,41,42,43,44,45] and supercritical fluids [24,30,32,46] and shows promise for understanding processes in the prenucleation state and predicting crystalline forms [46,47].

Bicalutamide (BCL) is an important compound used to treat prostate cancer [13,48]. Understanding its solid forms is crucial for improving solubility and bioavailability [15,49,50,51]. BCL’s pharmacological action involves blocking androgen receptors and is more tolerable than other antiandrogens [52,53,54]. However, its application may be limited by side effects such as gynecomastia and hematuria, highlighting the need for further research and development [55,56,57]. Two crystalline forms of BCL are known, including the stable form I (CSD Refcode: JAYCES01) in monoclinic symmetry and the metastable form II (CSD Refcode: JAYCES02) in triclinic symmetry [13,49,58]. The formation of these forms is determined by the molecule’s ‘open’ and ‘closed’ conformations, respectively. The type of conformational flexibility, where the change in dihedral angle leads to the molecule ‘folding’ or ‘unfolding,’ is quite common. It often characterises various solid forms and is observed in structures with cyclic fragments, often connected by extended aliphatic motifs [28,32,42,59,60,61,62]. This conformational flexibility is a critical factor in understanding the different solid forms of BCL and is a focus of our research.

The torsion angle C13–S–C12–C10 provides the main difference in the spatial structure of molecules within the unit cells of crystals, with values of −88.3(4)° and 72.5(4)° in forms I and II, respectively [49].

Various methods for obtaining polymorphic forms and co-crystals of BCL are presented in [63,64,65]. Perlovich et al. [13] conducted a comprehensive study of BCL in organic solvents and their mixtures to find and obtain new polymorphic and solvatomorph forms through crystallisation. A new BCL crystal solvate with dimethyl sulfoxide (DMSO) (CSD Refcode: FAHFIG) was discovered, with a melting point of 115.3 °C. The dissolution enthalpy of the crystal solvate exceeds 20 kJ/mol, whereas the values for the two polymorphic forms and the amorphous state are less than 10 kJ/mol. It is also noted that the DMSO crystal solvate is less stable than the polymorphic forms I and II, as it quickly loses solvent at room temperature. For other solvents, such as methanol, ethanol, 2-propanol, water, etc., recrystallisation led to the formation of the stable polymorphic form I. Approximately 50 different solvent combinations were used, and only the I polymorphic modification of BCL was obtained [13]. The polymorphic form II was achieved using the method presented in [49]. Additionally, ref. [66] discusses the monocrystalline X-ray structure of BCL (CSD Refcode: JAYCES), noting that the molecule’s conformation in the single crystal is similar to that observed in form I, except for the position of the hydroxyl group. This conformational rearrangement disrupts the intramolecular hydrogen bond between the hydroxyl and sulfonyl groups in the structure, altering the crystal packing.

The authors of [67] presented the structures of two BCL co-crystals with 4,4′-bipyridine (CSD Refcode: KIHZOR) (melting point 157–159 °C) and trans-1,2-bis(4-pyridyl) ethene (CSD Refcode: KIHZIL) (melting point 161–163 °C), respectively. The lengths of the hydrogen bonds O-H⋯N are 2.76 Å and 2.81 Å, and H-N⋯N are 3.50 Å and 3.12 Å for the first and second co-crystals, respectively. Each BCL molecule in the co-crystal forms a horseshoe-like structure similar to the polymorphic form I of pure BCL. The authors also noted that using the liquid-assisted grinding method, only one form was isolated for the co-crystals.

In [68], the co-crystallized X-ray structure of the androgen receptor bound to BCL (PDB ID: 1Z95) is reported. The authors of [69] conducted an extensive conformational search based on data from the CCDC (Cambridge Crystallographic Data Center) and PDB (Protein Data Bank) databases. They found that BCL molecules exist in 4 conformations in various forms with significant differences in torsion angle values and absolute/relative energies. During the optimisation of four structures taken as a basis using the Becke 3-Lee–Yang–Parr (B3LYP) methods [70,71] with basis sets 6-311+G(d,p), 6-311++G(d,p), 6-311++G(2df,3pd) 6-311++G(d,p), and RMP2/6-31+G(d,p) [72], 18 possible conformations were identified, whose geometric and energy characteristics are discussed in detail in [69]. Optimising structures in the gas phase using the B3LYP/6-31+G(d,p) method revealed significant differences in the torsion angle τ_1_(C13–S–C12–C10) values in the crystal and gas/solution. The authors also noted differences in the intermolecular hydrogen bonds of the sulfone with hydroxyl and amide groups when analysing crystalline structures.

The broad interest in modifying this compound is primarily driven by enhancing the solubility of the obtained forms. For instance, Perlovich et al. [13] noted that the obtained BCL crystal solvate with DMSO demonstrated a concentration increase up to 3.3 times higher than form I over the same period. A similar trend is observed in the solubility characteristics of the metastable form II and the amorphous state, with concentrations of ≈3.0 × 10^−7^ and 3.3 × 10^−7^ mol. frac., respectively, exceeding the analogous values for form I by more than two times. In summary, information on the spatial structure of bicalutamide molecules in saturated solutions is crucial for developing effective forms with specified properties.

## 2. Results and Discussion

### 2.1. Quantum-Chemical Calculations

The compound (((2RS)-4′-cyano-3-(4-fluorophenyl-sulfonyl)-2-hydroxy-2-methyl-3′-(trifluoromethyl)-propionate-anilide)) consists of two substituted cyclic fragments (A and B) and an aliphatic linker region C (see Figure 1). Fragment A is a substituted six-membered ring (benzonitrile) with a trifluoromethyl group in the meta-position at carbon C6, and fragment B is a fluorobenzene fragment of the molecule. The aliphatic region is a hetero-substituted carbon chain including amino, carbonyl, hydroxyl, and sulfonyl groups. The structural fragment –N(H)–C(O)– consists of two interconnected functional groups: a carbonyl (–C=O) and an amino group (–N(H)–).

In the crystalline state, BCL is a racemic mixture of two isomers, R and S [69,73], with chirality at carbon C10 (see Figure 1). While in the crystal form, BCL can exhibit two conformationally defined polymorphic forms (I and II), with synthesis methods and physicochemical characteristics described by Vega et al. [49]. Form I, unlike form II, is characterised by a moderate intramolecular hydrogen bond between the hydroxyl group at carbon C10 and the sulfonyl oxygen. Additionally, the same authors confirmed the existence of intra- and intermolecular π⋯π-stacking interactions in forms I and II, respectively.

BCL molecules in polymorphic forms primarily differ in the torsion angle τ_1_(C10–C12–S–C13), with values of −88.3° and 72.5° for forms I and II, respectively [49]. A conformational search within quantum chemical calculations showed that varying the positions of the cyclic fragments relative to each other, accompanied by changes in the τ_1_ angle (see Figure 1), results in many different conformers. While differences in other dihedral angles within the linker aliphatic fragment C of BCL molecules exist [69], such changes are not characteristic for identifying solid forms and are not discussed in this study.

As demonstrated in previous works [30,74], it should be noted that accounting for solvent effects using continuum models such as PCM in quantum chemical calculations can significantly reduce the selection of probable conformers. The potential errors due to the omission of solvent effects can significantly impact the interpretation of NOESY data and lead to inaccuracies in determining the proportions of conformer groups [29]. Therefore, we conducted the conformational search in the gas phase to ensure thorough research. The energy difference between BCL conformers 1–10 is 31.02 kJ/mol, while the differences between BCL-1 and BCL-2 and between BCL-1 and BCL-3 are 1.35 kJ/mol and 25.59 kJ/mol, respectively. The further conformational search was unnecessary, as the obtained structures were unlikely to exist in the solution. The values of the τ_1_ angles, relative conformer energies, hydrogen bond energies, distances, and angles, as well as images of the conformer structures, are provided in Table 1. Cartesian coordinates of the optimised geometries of conformers are given as Appendix A.

According to the quantum chemical calculations summarised in Table 1, BCL conformers are stabilised by weak intramolecular hydrogen bonds, with two present in each structure. Conformers BCL-1 to BCL-5 and BCL-9 and BCL-10 feature pairs of hydrogen bonds, N-H...OH and O-H...OS, with bond energies around −30 kJ/mol. The N-H...OH bond is also present in conformers BCL-6 and BCL-7, but the second hydrogen bond in these conformers is N-H...OS, with an energy of only −9 kJ/mol. The strongest hydrogen bonds N-H...OS and O-H...OC, with energies exceeding −40 kJ/mol, were observed in conformer BCL-8. Additionally, the relative orientation of A and B rings in BCL indicates that molecules can adopt “closed” (BCL-1, BCL-2, BCL-6, BCL-7) and “open” (BCL-3, BCL-4, BCL-5, BCL-8, BCL-9, BCL-10) conformations (see Figure 2), where the peripheral fluorine atoms in the cyclic fragments are aligned, with distances < 6 Å, or opposite with distances > 9 Å, respectively [75]. Similar conformational change trends are observed in structures forming different polymorphic forms, as confirmed by crystallographic analysis data [49]. Additionally, several studies [69,76,77] focused on BCL’s conformational state characterisation, which is consistent with the results presented here. However, within this study’s broad range of conformers, the τ_1_ dihedral angle values are unreliable for distinguishing “closed” and “open” conformers, as their values vary widely within each group. As will be shown later, BCL’s conformational flexibility is unique in that no single dihedral angle allows for a clear separation of the conformer set, complicating conformational analysis. Thus, internuclear distance values may be suitable for distinguishing structural motifs and calculating their proportionate distribution in the studied systems.

Furthermore, in our quest to understand the influence of the environment (CHCl_3_ and DMSO) on the distribution of conformers, we conducted quantum chemical calculations using the integral equation formalism variant of the polarisable continuum model (IEFPCM) [78,79]. These calculations led to the discovery of Gibbs solvation energy values (∆G_solv_) for each of the 10 conformers (see Figure 3a), a significant finding in our study. We also calculated the relative free energies of the conformers (∆G_CHCl3_ and ∆G_DMSO_), considering the environment’s influence (see Figure 3b).

The analysis of the obtained results indicated that the Gibbs solvation energies were −41.26 ± 6.53 kJ/mol and −58.44 ± 8.99 kJ/mol for the conformers of bicalutamide in CHCl_3_ and DMSO, respectively. These findings significantly impact our understanding of bicalutamide behaviour in different solvents. The values of the relative free energies of the conformers considering the influence of the medium (∆G_CHCl3_, ∆G_DMSO_) and those calculated in the gas phase (∆G_g_) are comparable (see Figure 3b). Similarly to ∆G_solv_, a decrease in energy values was observed for the BCL-6 and BCL-7 conformers in different solvents, indicating a weak influence of the medium on the energetic and structural characteristics of the bicalutamide conformers. As shown in the graphs, the contribution of Gibbs solvation energy is insufficient to redistribute the low-energy “closed” and high-energy “open” conformers, which opens up new avenues for further research.

The calculation of the Boltzmann factor (exp(−∆G/RT)) was conducted with utmost precision and did not yield results different from those in the gas phase (see Figure 4). According to quantum-chemical calculations, the likely conformers are BCL-1 and BCL-2, which belong to the “closed” type regardless of the medium. Therefore, the calculations of the conformer group fractions based on the NOESY experimental data relied on the structural data of the conformers obtained from the gas phase, ensuring the reliability of our findings.

### 2.2. Results of NMR Experiments

#### 2.2.1. NMR Spectrum Analysis

To calculate the proportions of BCL conformer groups, resonance signals in ^1^H NMR spectra (see Figure 5) were analysed in solvents of varying polarity (CDCl_3_ and DMSO-d_6_). The assignment of resonance signals was performed using one-dimensional (^13^C) (see Appendix A) and two-dimensional NMR approaches (^1^H-^13^C HSQC, ^1^H-^13^C HMBC, ^1^H-^1^H TOCSY with different mixing times) (see Appendix A). Note that ^13^C spectra of BCL in CDCl_3_ were not provided due to the low solubility of the research object and the resulting high signal-to-noise ratio, which prevents signal assignment. Nevertheless, the comprehensive analysis revealed correlations between carbon and hydrogen atoms through one (^1^H-^13^C HSQC) and more than one (^1^H-^13^C HMBC) chemical bonds, as well as correlations between hydrogen atoms in a continuous chain of chemical bonds (^1^H-^1^H TOCSY), forming a reliable basis for assigning resonance signals in ^1^H and cross-peaks in ^1^H-^1^H NOESY NMR spectra. All obtained information on the assignment of resonance signals in 1D and cross-peaks in 2D NMR spectra is provided in Appendix A.

Solvent changes lead to variations in chemical shift values of signals corresponding to hydrogen atoms in BCL molecules. In the high field region, there are two signals, one at 0 ppm corresponding to methyl groups of the internal standard tetramethylsilane present in the commercially available solvents. Additionally, a strong singlet signal is observed for the hydrogen atoms of the BCL methyl group (H11 (DMSO-d_6_) − s, δ = 1.43 ppm; H11 (CDCl_3_) − s, δ = 1.62 ppm). Signals for the CH_2_ group protons are found between 3 to 7 ppm (H12a (DMSO-d_6_) − d, ^2^J_12a-12b_ = 14.86 Hz, δ = 3.74 ppm; H12a (CDCl_3_) − d, ^2^J_12a-12b_ = 14.45 Hz, δ = 3.51 ppm; H12b (DMSO-d_6_) − d, ^2^J_12b-12a_ = 14.86 Hz, δ = 3.97 ppm; H12b (CDCl_3_) − d, ^2^J_12b-12a_ = 14.45 Hz, δ = 3.96 ppm), along with the hydroxyl proton signal (OH (DMSO-d_6_) − s, δ = 6.44 ppm; OH (CDCl_3_) − s, δ = 5.07 ppm). Signals for aromatic fragment protons are located in the 7 to 10 ppm range (H15/17 (DMSO-d_6_) − t, ^3^J_15/17-14/18_ = 8.78 Hz, δ = 7.37 ppm; H15/17 (CDCl_3_) − t, ^3^J_15/17-14/18_ = 8.38 Hz, δ = 7.21 ppm; H14/18 (DMSO-d_6_) − m, ^3^J_14/18-15/17_ = 13.90 Hz, ^5^J_14/18-12b/12a_ = 3.50 Hz, δ = 7.94 ppm; H14/18 (CDCl_3_) − m, ^3^J_14/18-15/17_ = 13.50 Hz, ^5^J_14/18-12b/12a_ = 3.75 Hz, δ = 7.91 ppm; H3 (DMSO-d_6_) − dd, ^3^J_3-4_ = 8.60 Hz, ^4^J_3-1_ = 1.35 Hz, δ = 8.24 ppm; H4 (DMSO-d_6_) − d, ^3^J_4-3_ = 8.60 Hz, δ = 8.09 ppm; H3/4 (CDCl_3_) − δ = 7.83 ppm; H1 (DMSO-d_6_) − d, ^4^J_1-3_ = 1.15 Hz, δ = 8.45 ppm; H1 (CDCl_3_) − s, δ = 8.01 ppm), along with NH group signals (NH (DMSO-d_6_) − s, δ = 10.41 ppm; NH (CDCl_3_) − s, δ = 9.09 ppm). Analysis of chemical shift values indicates that upon switching from CDCl_3_ to DMSO-d_6_, all signals, except H11, shift downfield, likely due to deshielding influenced by the phenyl fragment ring currents during conformational rearrangements. A similar trend was observed in arbidol conformational equilibria studies in varying polarity solvents [32].

A distinctive feature of ^1^H NMR spectra of BCL is the separation of diastereotopic protons CH_2_ group signals H12a and H12b near the molecule’s chiral centre [77,80,81], reflecting the presence of two BCL enantiomers. Also noteworthy are signals corresponding to water protons in the system (δ(H_2_O) = 1.62 ppm − CDCl_3_ and 3.39 ppm − DMSO-d_6_). Based on the isotopic purity of the solvents used (99.8 and 99.9 atom % D for CDCl_3_ and DMSO-d_6_, respectively) and the integral intensity values of hydrogen signals from H_2_O and the solvent, the water content in the studied systems is 0.09% and 0.006%, respectively. The chemical shift values of signals for characteristic atom groups in BCL structures are typical and align with literature data [82,83].

A comparison of the obtained resonance signal assignments in the ^1^H NMR spectra with results from [80,84], which provided similar data for BCL in D_2_O/CD_3_OD (8:2, *v*/*v*) and D_2_O containing 5% DMSO-d_6_, revealed that the CH_3_ group protons (H11) also show intense signals in the high field region (δ ≈ 1.5 ppm). Moving along the chemical shift axis to the downfield region, signals for CH_2_ group protons (H12a, H12b − δ ≈ 3.9 ppm) appear, with the reported δ value as the centred signal for the AB system. The hydroxyl group proton signal (OH) is found at δ ≈ 6.5 ppm, according to [84]. In contrast, the amino group proton (NH) occupies the far-left position in the downfield region, which is consistent with the same data. Other signals related to the aromatic fragments of BCL molecules are similarly positioned between 7.2 ppm and 8.4 ppm. The protons of the fluorobenzene fragment exhibit identical chemical shift values for ortho- (δ ≈ 7.9 ppm) and meta- (δ ≈ 7.2 ppm) positions. The study [81] performed assignments of ^1^H NMR signals in BCL spectra in DMSO-d_6_, recorded on a Varian Gemini 2000, 200 MHz spectrometer at 25 °C. This study’s signal sequence and chemical shift values align with our results, confirming the data, although the chemical shifts and spectral resolution differ. The chemical shift values of resonance signals in the ^1^H and ^13^C NMR spectra of pure BCL in CDCl_3_, obtained using two-dimensional NMR approaches, have not been previously reported in the literature.

The analysis of spin-spin coupling constants (SSCC) (see Figure 5) further confirms the correct assignment of resonance signals and aligns with literature data on BCL structure characteristics using NMR methods [81]. Results indicate a decrease in SSCC values for corresponding signals when transitioning from DMSO-d_6_ to CDCl_3_ within 1 Hz, expected with solvent changes and potentially related to molecular geometry alterations during conformational rearrangements due to solvation effects [85,86,87]. The obtained SSCC values are also consistent with literature data [81,88], with the constant values for the doublet signals of protons H12a and H12b being 14.78 Hz [81], while our values are ^2^J_12a-12b_ = 14.86 Hz and 14.45 Hz in DMSO-d_6_ and CDCl_3_, respectively. This trend is also observed for the trifluoromethyl-substituted benzonitrile fragment protons, where the constant value for the doublet of proton H4 is 8.44 Hz, according to the literature [81]. In comparison, our value from the ^1^H BCL spectrum in DMSO-d_6_ is ^3^J_4-3_ = 8.60 Hz. Constants ^4^J_3-1_ = 1.35 Hz and ^4^J_1-3_ = 1.15 Hz from BCL spectra analysis in DMSO-d_6_ also agree with the literature results at 1.56 Hz. However, data on the fine structure of signals H14/18 and H15/17 are unavailable in the literature.

#### 2.2.2. Reference-Free NOE NMR Analysis

Analysis of NOESY spectra recorded in CDCl_3_ and DMSO-d_6_ (see Figure 6) revealed 11 and 15 pairs of cross-peaks related to hydrogen atoms within 5–6 Å in BCL molecules. Most cross-peaks are familiar to BCL regardless of the solvent. The observed differences are due to signal overlap, primarily in the downfield region where protons of the cyclic fragment with a trifluoromethyl radical are located, or low signal intensity, which may be related to both BCL concentration in CDCl_3_ and changes in rotational correlation times with solvent changes [89].

To determine probable BCL conformers in different environments, an attempt was made to use the approach proposed by Roberto R. Gil and co-authors in [90]. We have previously applied reference-free NOE NMR analysis to find probable conformers of 1,5-diaryl-3-oxo-1,4-pentadienes [34]. However, satisfactory results were not achieved due to the large set of probable conformers and high conformational lability. For using reference-free NOE NMR analysis, normalised integral intensities of all cross-peaks observed in NOESY spectra and cross-relaxation rates for proton pairs were calculated using the PANIC model (Peak Amplitude Normalization for Improved Cross-relaxation) [91,92,93,94], within the IRA (initial rate approximation) framework [95,96] (see Appendix A). The magnetisation exchange between two adjacent proton spins within the molecule is primarily driven by dipole cross-relaxation. Thus, cross-relaxation rates *σ*_exp_ (see Equation (1)) [37,97,98,99,100] accurately measure this process.
(1)$$\sigma_{\exp}=\frac{\hbar^{2}\mu_{0}^{2}\gamma^{4}}{40\pi^{2}r_{\exp}^{6}}\left\{-1+\frac{6}{1+4\omega_{0}^{2}\tau_{c}^{2}}\right\}\to \sigma_{\exp}∼\frac{1}{{r_{\exp}}^{6}}$$
where *ħ* is the reduced Planck constant, *µ*_0_ is the magnetic permeability, *ω*_0_ is the Larmor frequency, and *γ* is the gyromagnetic ratio.

Thus, as seen from Equation (1), the cross-relaxation rate determines the internuclear distance *r*_exp_. Additionally, as noted by Lee and Krishna [101], the cross-relaxation rate is a weighted average over all conformers in the studied system, which can be expressed by Equation (2).
(2)$$\sigma_{\exp}=\sum_{i}\sigma_{i}x_{i}$$
where *σ_i_* is the cross-relaxation rate in conformer *i*, and *x_i_* is the relative proportion of that conformer.

This study did not determine the cross-relaxation rate for the H14/18-H15/17 distance in CDCl_3_ due to significant T1 noise contributions to integral intensity values. Similarly, values for H3-H4 and H12a-H12b distances in DMSO-d_6_ were not determined due to the proximity of cross and diagonal peaks, resulting in significant errors in integral intensity determination. Cross-relaxation rates for other distances are provided in Appendix A.

For reference-free NOE NMR analysis, corresponding internuclear distances with observed NOESY cross-peaks were calculated, considering intramolecular mobility for each of the 10 conformers obtained from quantum chemical calculations. Previously, the Tropp model was most effective in accounting for intramolecular mobility. However, ref. [42] proposed semi-empirical coefficients for spherical harmonics, allowing precise consideration of methyl group mobility and increasing the accuracy of determining internuclear distances involving them. Sternberg and Witter [102] showed that an incorrect averaging model could lead to erroneous NOESY data interpretation, ultimately complicating accurate internuclear distance assessment. This conclusion underscores the importance of proper averaging model application, suggesting more complex models.

Consequently, internuclear distances corresponding to NOESY cross-peaks were determined using three averaging models, considering intramolecular lability characterised by rotational correlation times and group movement speeds—slow (>100 ps), medium (50–100 ps) and fast (<50 ps) [93].

Typically, for calculating distances involving hydrogen atoms in benzene rings, the averaging model in Equation (3) is used [103,104]:(3)$$r_{calc}={\left[\frac{1}{n_{I}n_{S}}\sum_{i}\frac{1}{r_{i}^{6}}\right]}^{-\frac{1}{6}}$$

To determine distances between fragments with medium movement speeds, such as between CH_2_ group protons, Equation (4) is used:(4)$$r_{calc}={\left[{\left(\frac{1}{n_{I}n_{S}}\sum_{i}\frac{1}{r_{i}^{3}}\right)}^{2}\right]}^{-\frac{1}{6}}$$
where *r_calc_* is the average internuclear distance obtained from conformer structures, *n_I_* and *n_S_*, are the number of equivalent atoms in groups *I* and *S*, and *r_i_* is the distance between atoms in the considered groups.

Equation (5) determines distances between fragments with fast movement types, such as methyl groups. This model includes spherical harmonics and their coefficients, previously mentioned [42], to account for complex movement types in the considered fragments accurately.
(5)$$r_{calc}={\left[\frac{1}{5}\sum_{k=-2}^{2}\left|\frac{1}{3}\sum_{i=1}^{3}\frac{Y_{2k}\theta_{mol}^{i},\varphi_{mol}^{i}}{r_{i}^{3}}\right|\right]}^{-\frac{1}{6}}$$
where *θ_mol_* and *φ_mol_* are the polar angles of the internuclear vector in the molecular frame of reference, and *Y*_2*k*_ are second-order spherical harmonics.

The DIST_ACCESS program (RU 2020618574), included in the Unified Registry of Russian Programs for Electronic Computers and Databases, was used for the calculations. Data on internuclear distances and cross-relaxation rates were used to construct graphs, presented in Figure 7. The obtained dependencies were approximated using the exponential model described in [90]. Graphs were created for the ten BCL conformers in CDCl_3_ (see Appendix A) and DMSO-d_6_ (see Appendix A).

The graphs constructed for the first conformer show that experimentally obtained cross-relaxation rate values for BCL internuclear distances in CDCl_3_ better agree with corresponding internuclear distances than those obtained from NOESY spectra analysis in DMSO-d_6_. Determination coefficients of 0.939 and 0.347 confirm this observation for data obtained in CDCl_3_ and DMSO-d_6_, respectively. Similar results were obtained when analysing cross-relaxation rate dependencies on corresponding internuclear distances in other BCL conformers.

The joint analysis results of determination coefficient values are presented as diagrams in Figure 8. Despite the low dispersion of analysed values observed for the CDCl_3_ system, the obtained data do not unequivocally identify the predominant conformer of the studied object due to the close determination coefficient values. This proximity may be explained by the significant conformational flexibility of the BCL molecule in CDCl_3_, as in previously considered cases [34]. In contrast, the DMSO-d_6_ results show the opposite trend—data poorly fitting the model may show negative determination coefficients for nonlinear functions. It is noteworthy that when identifying probable BCL structures in CDCl_3_ based on determination coefficients, conformers BCL-2, BCL-3, BCL-5, BCL-6, BCL-8, and BCL-10 exhibit higher values, whereas analysing DMSO-d_6_ spectral data identifies probable conformers as BCL-1, BCL-4, BCL-7, and BCL-9. We acknowledge that this situation is likely coincidental, requiring further confirmation, and cannot be used for definitive conclusions.

The analysis shows that the exponential dependence of the cross-relaxation rate on the corresponding internuclear distance is only sometimes maintained. For conformationally flexible molecules, an alternative approach based on the quantitative determination of experimental internuclear distance values dependent on molecular conformation is required [32,36,42,105].

#### 2.2.3. Method Based on Determining Experimental Values of Internuclear Distances

The previously obtained results on the determination of cross-relaxation rates and calculated internuclear distances were utilised to construct graphs (see Figure 9) based on data for all BCL conformers to select appropriate conformation-dependent (r_exp_) and reference (r_0_) distances.

As seen from the graphs, the distances H12a-H12b and H15/17-H14/18 in chloroform and DMSO-d_6_, respectively, undergo minimal changes in value depending on the conformation of BCL molecules (see Table 2), making them the only candidates for use as reference distances. Other distances can be conformation-dependent but are unsuitable for precise quantitative assessment of the proportions of “open” and “closed” BCL conformer groups. Some cross-peaks observed in the NOESY spectrum of BCL in DMSO-d_6_ correspond to distances that exceed the experimental sensitivity limit for certain conformers (H14/18-NH, OH-H14/18, H11-H14/18). Using these distances in calculations will inevitably lead to significant errors in the obtained results, as previously demonstrated in [42]. Additionally, the distances H1-NH, H3/4-NH (CDCl_3_), and H3-NH (DMSO-d_6_) determine the conformational flexibility of the molecule but relate to the positional change of the trifluoromethyl-substituted cyclic fragment relative to the aliphatic structure and do not allow the separation of BCL conformers into “open” and “closed” groups. Therefore, a comprehensive analysis established that the H12b-H14/18 distance could be conformation-determining. The calculated distance values range from 2.97 Å for conformer 10 to 4.33 Å for conformer 1 (see Table 2). These distance values allow the identification of two groups of conformers corresponding to the “open” and “closed” types with average values of 3.35 Å and 4.21 Å, respectively. Notably, the H12b-H14/18 distances obtained using Equation (3) show the best results and allow accurate comparison with NOESY experimental data despite the participation of hydrogen atoms (H12b) in the CH_2_ group. The developed structure of intramolecular hydrogen bonds in different BCL conformers likely restricts the mobility of CH_2_ group protons, necessitating the use of the slow-motion averaging model.

To determine the values of conformation-determining distances based on NOESY experimental data within the isolated spin pair approximation (ISPA) model, cross-relaxation rates for reference and target distances were used (see Equation (6)) [106,107,108].
(6)$$r_{\exp}=r_{0}{\left(\frac{\sigma_{0}}{\sigma_{\exp}}\right)}^{\frac{1}{6}}$$
where *r*_exp_ is the target conformation-determining distance, *r*_0_ is the reference distance based on quantum chemical calculations, *σ*_0_ and *σ*_exp_ are the cross-relaxation rates for reference and conformation-determining distances.

The cross-relaxation rates for reference and conformation-determining distances, calculated using the IRA model, were (2.46 ± 0.15) × 10^−1^ s^−1^ and (1.29 ± 0.09) × 10^−2^ s^−1^ for H12a-H12b in CDCl_3_ and H15/17-H14/18 in DMSO-d_6_, respectively. For the H12b-H14/18 conformation-determining distance, the cross-relaxation rates were (2.35 ± 0.07) × 10^−3^ s^−1^ and (3.12 ± 0.13) × 10^−2^ s^−1^, derived from NOESY data in the same solvents (see Figure 10). The increase in cross-relaxation rates when transitioning from CDCl_3_ to DMSO-d_6_ is expectedly accompanied by a decrease in conformation-determining distances, which are 3.86 Å and 3.55 Å in CDCl_3_ and DMSO-d_6_, respectively.

The experimental values of the H12b-H14/18 distance fall within the calculated range, allowing for the determination of the proportions of “open” and “closed” conformers within the two-state exchange model (see Equation (7)):(7)$$r_{\exp}=\sqrt[{6}]{\frac{r_{open}^{6}\times r_{close}^{6}}{P_{open}r_{close}^{6}+P_{close}r_{open}^{6}}}\to P_{open}=\frac{r_{open}^{6}(r_{close}^{6}-r_{\exp}^{6})}{r_{\exp}^{6}(r_{close}^{6}-r_{open}^{6})}$$
where *r*_exp_ is the experimental conformation-determining distance H12b-H14/18, *r_open_* and *r_close_* are the calculated distances for “open” and “closed” conformer groups, *P_open_* and *P_close_* are the proportions of “open” and “closed” conformer groups.

The methodology for determining conformer group proportions is described in previous works [34,105,109]. It can be visualised as a graph of probable experimental distance values (*r*_exp_) within the range of calculated values from the proportion of the conformer group (open or close). In the case of a two-state exchange (*P_open_*) ↔ (*P_close_*), the dependence represents a curve *r*_exp_ = f(*P_open_*) (see Figure 11). It was determined that the proportions of open-type conformers in CDCl_3_ and DMSO-d_6_ are 22.7 ± 3.7% and 59.8 ± 6.2%, respectively, while the proportions of closed-type conformers are 77.3 ± 3.7% and 40.2 ± 6.2%, respectively. The observed geometric changes in BCL molecules when transitioning to a more polar solvent are confirmed by data from [110], where the authors demonstrated the influence of the medium on the characteristics of the conformers of the study object using the PCM method. The distributions of conformer group fractions obtained differ from the results of the Boltzmann factor analysis (refer to Figure 4), indicating that quantitative information about the conformer structure parameters (inter-nuclear distances) is more valid than their calculated energy values. The discrepancies between the quantum-chemical calculation results and the NOESY experiment stem from the inability of the PCM models to accurately consider the impact of intermolecular interactions, including hydrogen bonds and π-π interactions. As we have found, these interactions are the driving force behind the intriguing redistribution of the fractions of “closed” and “open” conformer groups of bicalutamide.

Additionally, the results align with the data from the Perlovich group [13], where a BCL-DMSO solvate was obtained and registered in the Cambridge Crystallographic Data Centre under number 879630 (CSD Refcode: FAHFIG). The analysis of the BCL molecule structure within the solvate (see Figure 12) showed that the H12b-H14/18 distance is 3.16 Å, corresponding to the “open” conformation, where peripheral fluorine atoms are oppositely directed with a distance between them >9 Å. Furthermore, [67] discusses the role of numerous hydrogen bond synthons in forming various BCL co-crystals, in which the molecules predominantly adopt the “open” conformation.

To identify potential similarities in the structural motifs of BCL molecules in solution and solid form, root mean square deviation (RMSD) values were calculated using ChemCraft V.1.8 software (see Appendix A). As noted in [111], RMSD calculation remains the fastest way to measure the structural similarity of molecules, which is particularly important for large datasets. RMSD characterises the average displacement between structures and can be calculated using Equation (8):(8)$$RMSD=\sqrt{\frac{\sum_{i=1}^{N}d_{i}^{2}}{N}}$$
where *d_i_* is the distance between the *i*-th atom in two aligned structures, and *N* is the total number of equivalent atoms.

Known method limitations, such as size constraints of studied molecules [112], the requirement for identical chemical composition of molecular structures [111], and the peculiarities of comparing molecules with different conformational flexibility [113,114], do not affect the calculations conducted in this study. However, Appendix A (blue area) shows that the comparison of molecular structures obtained from quantum chemical calculations demonstrates comparably high RMSD values (≈2.8 Å). Nonetheless, for specific pairs, BCL-1–BCL-7 and BCL-2–BCL-6, RMSD values are 0.78 Å and 0.74 Å, respectively, indicating high degrees of similarity. These conformers are classified as “closed” types. The remaining structures, classified as “open” type conformers, exhibit higher variability in molecular fragment orientations, reflected in RMSD values. Subsequently, RMSD values were calculated by comparing structures obtained from quantum chemical calculations and literature data (see Appendix A) presented in CCDC and PDB databases for two BCL polymorphs (CSD Refcode: JAYCES01 and JAYCES02) [49], DMSO solvate (CSD Refcode: FAHFIG) [13], co-crystals (CSD Refcode: KIHZOR and KIHZIL) [67], bioactive form (PDB ID: 1Z95) [68], and single-crystal X-ray structure (CSD Refcode: JAYCES) [66] (see Appendix A). Appendix A (orange area) shows that the obtained RMSD values (≈2.7 Å) demonstrate significant differences in the analysed structures. The closest values are observed for BCL molecules in the co-crystal with 4,4′-Bipyridine (CSD Refcode: KIHZOR) and conformer BCL-1 (RMSD = 0.91 Å), classified as “closed” type. Regarding comparing structures obtained from databases, common motifs were not identified, with RMSD values ≈ of 1.7 Å (see Appendix A—green area).

Additionally, for detailed analysis, the values of seven dihedral angles determining the main conformational changes in the molecular structure and the values of internuclear distances defining potential intramolecular hydrogen bonds were calculated (see Appendix A). The data analysis showed that none of the dihedral angles and the set of intramolecular hydrogen bonds are characteristic for identifying common motifs between conformers observed in solid forms and gas phases. Nevertheless, the H12b-H14/18 distance values exhibit similar trends in “open” and “closed” conformers obtained from databases (see Table 3). In this case, the distance is a crucial characteristic for identifying conformer types determined in experiments and during the analysis of calculated data.

The table shows that using structures obtained from the CCDC and PDB databases allows for identifying “open” and “closed” bicalutamide conformers. The H12b-H14/18 distance values (*r_open_* and *r_close_*) are characteristic for these groups, at 3.38 Å and 4.19 Å, respectively. However, the previously used reference distances H12a-H12b and H14/18-H15/17 differ significantly. Using quantum chemical calculation data, these distances were established at 1.78 Å and 2.80 Å, respectively. However, Table 3 data show a decrease in H12a-H12b and H14/18-H15/17 distances by 0.19 Å and 0.18 Å, respectively. An attempt was made to calculate the conformer group proportions based on NOESY experimental data and seven structures obtained from the CCDC and PDB databases. The existing differences alter the experimental H12b-H14/18 distances (*r*_exp_) determined using the ISPA method (see Equation (6)), which are 3.45 Å and 3.31 Å for BCL in CDCl_3_ and DMSO-d_6_, respectively. Using the obtained experimental distance values in the two-state exchange equation (see Equation (7)), it was found that the proportions of “open” and “closed” BCL conformer groups in CDCl_3_ are 16.4% and 83.6%, respectively. In the case of DMSO-d_6_, Equation (7) has no solutions, as the experimental distance value does not fall within the range of calculated values obtained from the considered conformer structures, and the “open” conformer group proportion would exceed 100%. As noted by the authors of [69], the discussed structures obtained from solid BCL forms are unlikely in the solution phase, and their optimisation using a broad range of quantum chemical methods results in structures significantly different from the original ones. The same authors note that the bioactive conformation of BCL in the co-crystallized X-ray structure of the androgen receptor (PDB ID: 1Z95) has an entirely different molecular conformation compared to any of the structures described in the solid state, which is also reflected in the H12a-H12b and H14/18-H15/17 distances shown in Table 3. This case underscores the importance of using optimised structures to interpret NOESY experimental data accurately. These are a few tools for identifying common structural motifs in small molecules with high conformational flexibility and determining conformer group proportions.

## 3. Materials and Methods

This study utilised commercially available compounds from Sigma-Aldrich Rus (Moscow, Russia): bicalutamide (C_18_H_14_F_4_N_2_O_4_S-(S)-N-(4-Cyano-3-(trifluoromethyl)phenyl)-3-((R)-(4-fluorophenyl)sulfinyl)-2-hydroxy-2-methylpropanamide; CAS: 90357-06-5); chloroform-d_1_ (CDCl_3_; CAS: 865-49-6); and anhydrous dimethyl sulfoxide-d_6_ (DMSO-d_6_; CAS: 2206-27-1). Solutions of BCL in CDCl_3_ and DMSO-d_6_ for NMR experiments were prepared in standard NMR tubes (Wilmad, O.D. = 5 mm) without further purification of the components. NMR spectra were recorded on a Bruker Avance III 500 MHz spectrometer operating (Bruker, Karlsruhe, Germany) at 500.17 MHz and 125.77 MHz for ^1^H and ^13^C nuclei, respectively. Temperature stabilisation (25 °C) was achieved using a Bruker BVT-2000 heating unit and a Bruker BCU 05 cooling accessory (Bruker, Karlsruhe, Germany), with a temperature maintenance accuracy of ±0.1 °C. Chemical shift values in the NMR spectra were determined relative to the internal standard TMS included in the solvents used. For optimising the duration of 2D NMR experiments, the non-uniform sampling (NUS) method was employed [115,116,117,118,119].

^1^H NMR spectra were recorded using the standard pulse program “zg” in the TopSpin 3.6.1 software package. Spectra were recorded over a frequency range of 13.2 ppm, with a relaxation delay of 3 s and 32,768 data points. ^13^C NMR spectra were recorded using the pulse program “zgpg30” with proton decoupling during the relaxation delay to enhance signal intensity. Applying 30° radiofrequency electromagnetic pulses instead of 90° ensured rapid pulse generation and system relaxation. The ^13^C NMR spectra were recorded over a range of 237 ppm, with a relaxation delay of 1 s and 65,536 data points.

^1^H-^13^C HSQC spectra were recorded using the “hsqcedetgp” pulse program [120,121,122,123], which allows differentiation of CH, CH_3_, and CH_2_ group cross-peaks in molecular structures due to phase sensitivity. The spectral range was 14 ppm (F2–^1^H) and 290 ppm (F1–^13^C), with a relaxation delay of 1 s and data points of 2048 and 256 along the F2 and F1 axes, respectively. ^1^H-^13^C HMBC spectra were recorded using the “hmbcgpndqf” pulse program to observe long-range carbon-hydrogen correlations [124]. The spectral range was 20 ppm (F2–^1^H) and 290 ppm (F1–^13^C), with a relaxation delay of 1 s and data points of 2048 and 256 along the F2 and F1 axes, respectively.

Homogeneous ^1^H-^1^H TOCSY spectra were recorded using the “mlevgpphprzf” pulse program, based on Hartmann-Hahn homogeneous transfer using the MLEV-17 sequence, with a spin-lock parameter of 20 ms and 100 ms to observe short- and long-range proton-proton interactions in continuous chemical bond chains [125,126]. The spectral range was 15.4 ppm along the F2 and F1 axes, with a relaxation delay of 2 s and data points of 2048 and 256 along the F2 and F1 axes, respectively.

^1^H-^1^H NOESY spectra were recorded using the “noesygpphpp” pulse program [127,128]. The spectral range was 15.4 ppm along the F2 and F1 axes, with a relaxation delay of 1.3 s and data points of 2048 and 256 along the F2 and F1 axes, respectively.

Initial conformer configurations of BCL were obtained using the Gabedit V. 2.5.2 software package [129] with the Amber force field [130] and molecular dynamics method at 1000 K, followed by optimisation using the openmopac V. 22.11 package via the PM6 method [131]. Several attempts were made to generate a set of conformers based on different initial structures. The obtained structures were then optimised in the Gaussian V. 16 package using the Austin–Frisch–Petersson (APFD) functional [132,133] and the 6-311++g(2d,2p) basis set [134]. The selection of the functional is based on its specialisation for conformational searches relevant to small molecules containing cyclic fragments in their structure [93,95]. Additionally, including diffuse and polarisation functions in the basis set improves the description of electron density and spatial structure of small molecules [96]. A total of 10 conformer structures with different dihedral angles and energies were optimised. The quantum theory of atoms in molecules (QTAIM) [135,136,137] was employed to estimate the strength of various hydrogen bonds. The electron and potential energy density at critical points are proportional to the bond energy. We utilised the correlation proposed by Espinosa et al. [98] to estimate the bond energy. Furthermore, Gibbs solvation energies and relative free energies of the conformers were calculated considering the environment within the IEFPCM model [78,79,138]. The Gibbs solvation energies were calculated as the difference in total electronic energies of the conformers in the gas phase and the solvent environment [138].

## 4. Conclusions

Our comprehensive bicalutamide (BCL) conformers analysis, employing quantum chemical calculations and NMR spectroscopy, has unveiled novel insights into this compound’s structural and energetic characteristics. BCL’s conformational flexibility, transitioning between ‘closed’ and ‘open’ conformations under intramolecular hydrogen bonding and π-π stacking interactions, is a significant finding. The energy differences between various conformers, primarily determined by the torsion angle τ_1_(C10–C12–S–C13), underscore the delicate balance between stability and structural variability. Quantum chemical calculations have further clarified the relative energies and hydrogen bond strengths across ten conformers, confirming the role of weak intramolecular hydrogen bonds in stabilising ‘closed’ conformations. The NOESY spectra analysis has highlighted the impact of solvent polarity on BCL conformations. The identification of conformation-determining distances, particularly H12b-H14/18, has enabled a quantitative assessment of ‘open’ and ‘closed’ conformer proportions, revealing a significant solvent effect, with higher proportions of ‘open’ conformers in more polar solvents like DMSO.

The results of this study underscore the crucial role of intramolecular interactions and solvent effects on BCL’s conformational properties. This study highlights the need for further research into BCL’s conformational behaviour in various environments and offers a promising avenue for enhancing its pharmaceutical applications and efficacy.

## Figures and Tables

**Figure 1 ijms-25-08254-f001:**
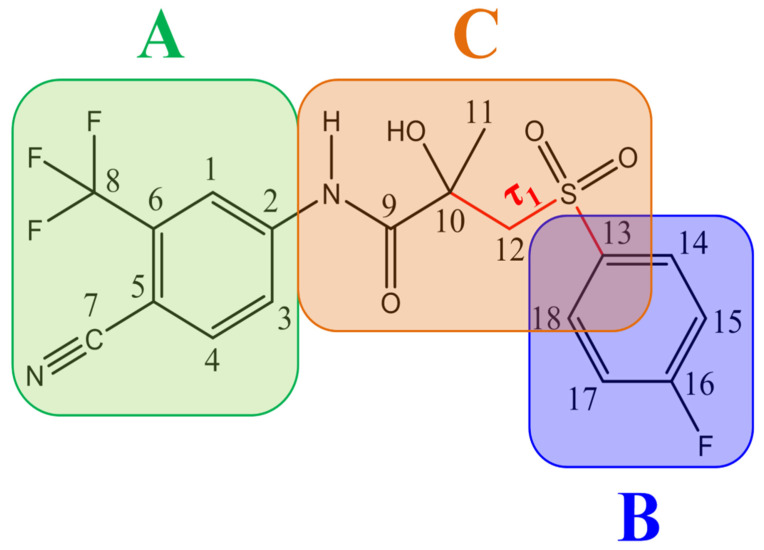
The structure of the BCL molecule with atom numbering is used to interpret the NMR spectrum and designate the angle τ_1_(C10–C12–S–C13) (red line), fragment A—benzonitrile with a trifluoromethyl group; fragment B—fluorobenzene; fragment C—aliphatic region is a hetero-substituted carbon chain.

**Figure 2 ijms-25-08254-f002:**
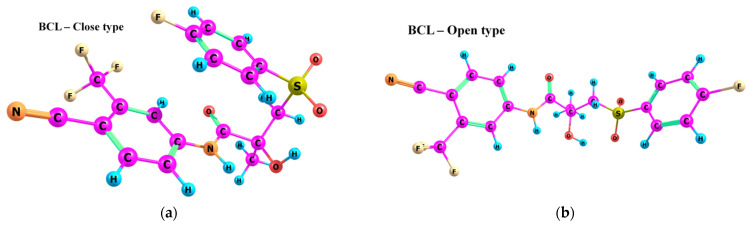
The molecular structure of “closed” (**a**) and “open” (**b**) BCL conformations, excluding atom numbering for simplicity. The structures were created with ChemCraft 1.8.

**Figure 3 ijms-25-08254-f003:**
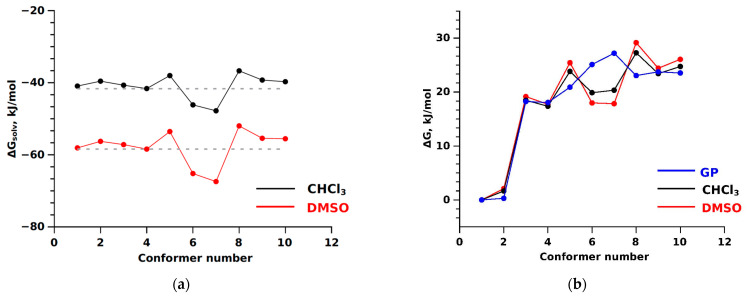
(**a**) Gibbs solvation energies for bicalutamide conformers obtained from quantum chemical calculations using the IEFPCM model, accounting for the influence of the solvents CHCl_3_ (black line) and DMSO (red line), average Gibbs solvation energy (dashed line); (**b**) relative free energies of bicalutamide conformers obtained from quantum chemical calculations in the gas phase (blue line) and considering the solvent effects (black and red lines).

**Figure 4 ijms-25-08254-f004:**
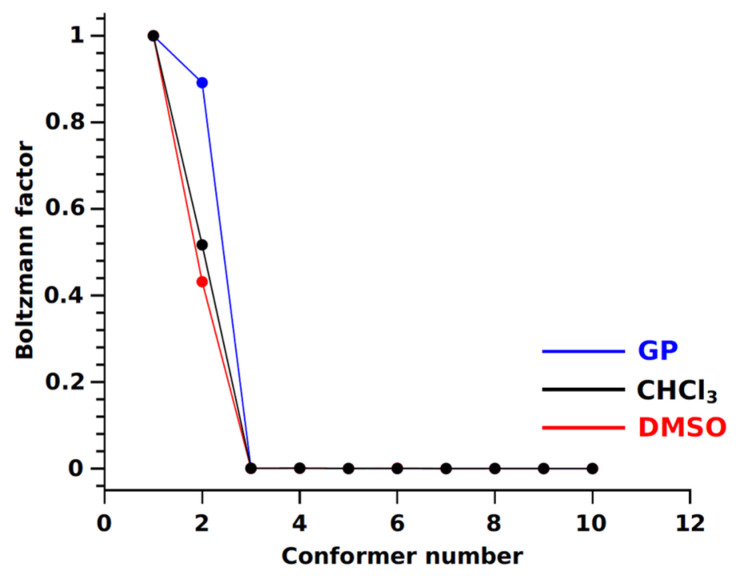
The Boltzmann factor for bicalutamide conformers was obtained from quantum chemical calculations performed in the gas phase (blue line) and with the inclusion of solvents CHCl_3_ (black line) and DMSO (red line).

**Figure 5 ijms-25-08254-f005:**
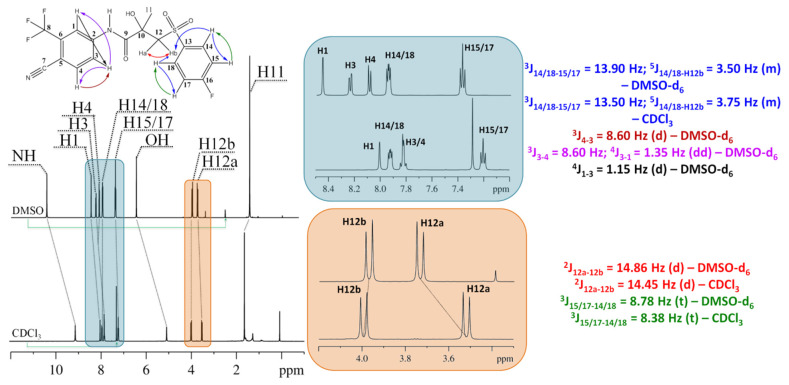
^1^H NMR spectra of BCL in DMSO-d_6_ and CDCl_3_ within analyses of couplings constants and molecular structure assignments.

**Figure 6 ijms-25-08254-f006:**
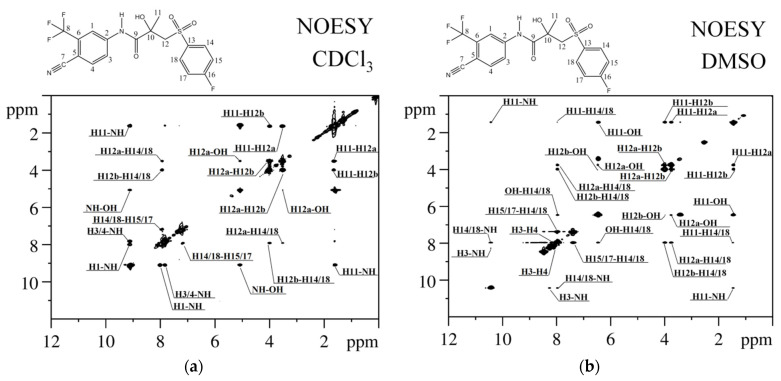
^1^H-^1^H NOESY NMR spectra of BCL in CDCl_3_ (**a**) and DMSO-d_6_ (**b**).

**Figure 7 ijms-25-08254-f007:**
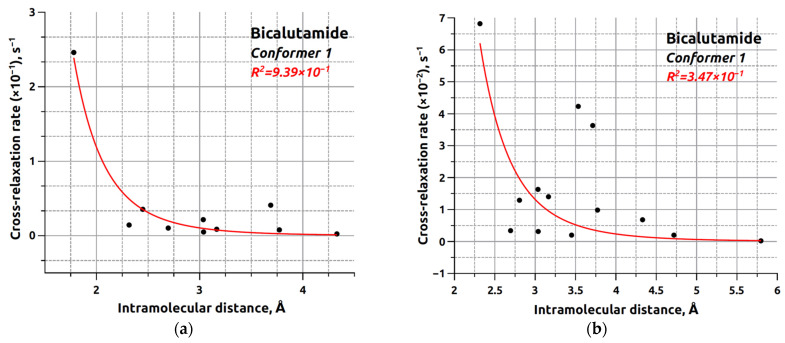
Dependence of the cross-relaxation rate on the internuclear distance for one of the conformers for bicalutamide in CDCl_3_ (**a**) and DMSO-d_6_ (**b**). The red line indicates the approximating curve 1/r^6^.

**Figure 8 ijms-25-08254-f008:**
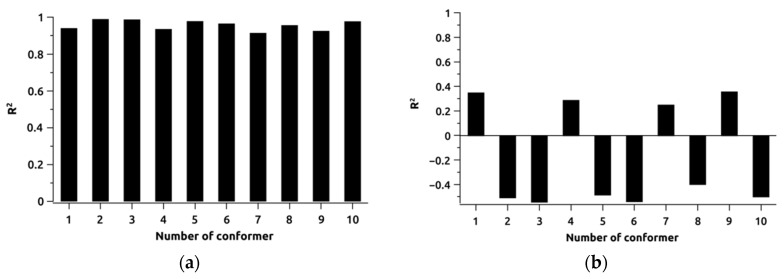
Distribution of coefficients of determination obtained for each conformer of bicalutamide in CDCl_3_ (**a**) and DMSO-d_6_ (**b**).

**Figure 9 ijms-25-08254-f009:**
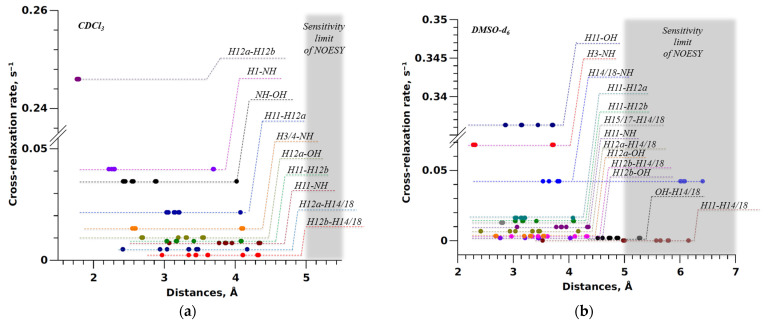
Dependence between the cross-relaxation rate determined from NOESY experiments in CDCl_3_ (**a**) and DMSO-d_6_ (**b**) and the internuclear distances in BCL conformers.

**Figure 10 ijms-25-08254-f010:**
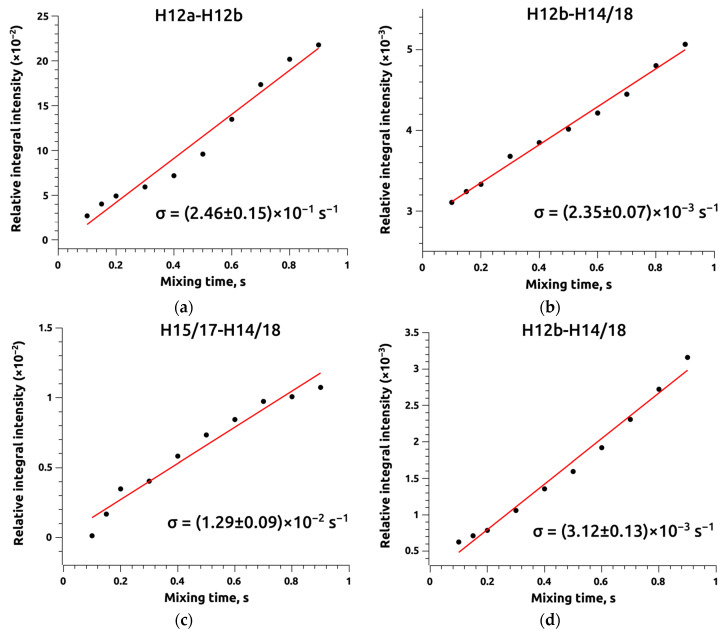
Average integral intensity of reference distances H12a-H12b (**a**), H15/17-H14/18 (**c**), and experimental distances H12b-H14/18 (**b**,**d**) derived from NOESY spectral analysis for bicalutamide in CDCl_3_ (**a**,**b**) and DMSO-d_6_ (**c**,**d**).

**Figure 11 ijms-25-08254-f011:**
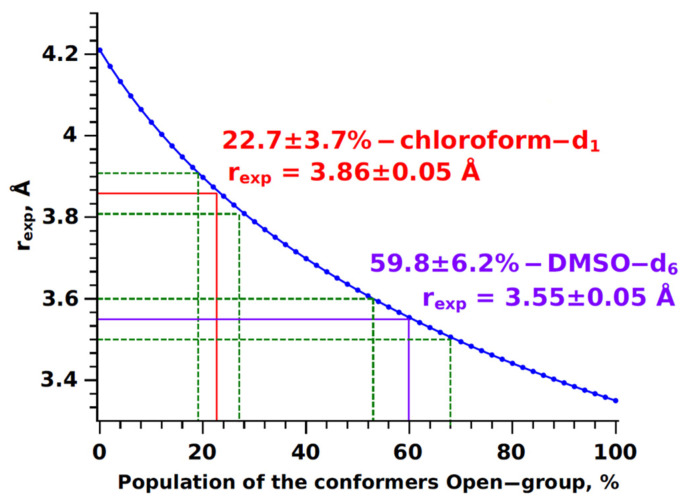
Graphs showing the change in the proportion of “open” type BCL conformers depending on the H12b-H14/18 internuclear distance (blue line). Red lines indicate results for CDCl_3_, purple for DMSO-d_6_, and green dashed lines show the error range for determining experimental distance and conformer group proportion.

**Figure 12 ijms-25-08254-f012:**
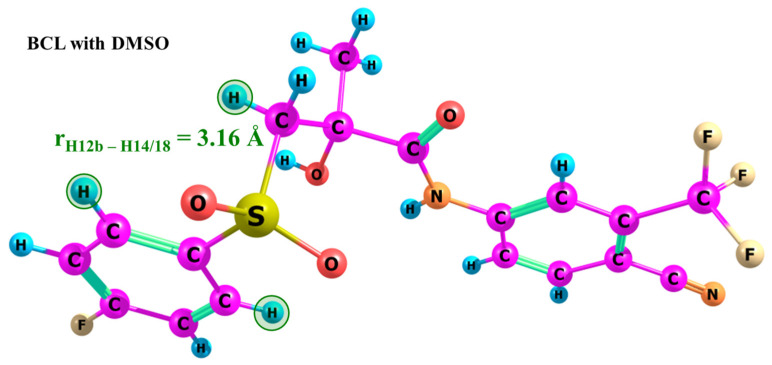
Structure of the BCL molecule within the DMSO solvate. Hydrogen atoms forming the experimental H12b-H14/18 distance are highlighted in green.

**Table 1 ijms-25-08254-t001:** Relative conformer energies, energies, distances and angles of hydrogen bonds present in the structures of bicalutamide conformers.

No	∆E, kJ/mol	HB Type	E_HB_, kJ/mol	R(X...Y), Å	R(H...Y), Å	∠(X-H...Y), °	Structure
1	0.00	N-H...OH	−28.47	2.607	2.013	114.95	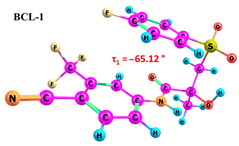
		O-H...OS	−38.32	2.726	1.818	154.04	
2	1.35	N-H...OH	−28.99	2.603	2.005	115.25	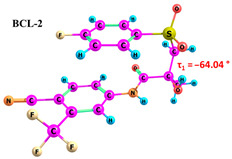
		O-H...OS	−38.56	2.726	1.816	154.36	
3	25.59	N-H...OH	−27.59	2.615	2.025	114.77	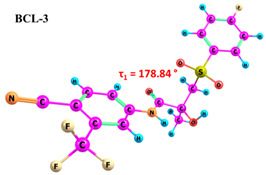
		O-H...OS	−33.99	2.767	1.869	152.55	
4	26.01	N-H...OH	−27.50	2.616	2.026	114.77	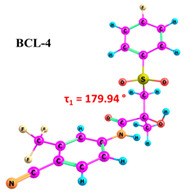
		O-H...OS	−33.54	2.772	1.874	152.61	
5	29.71	N-H...OH	−30.49	2.533	1.949	113.81	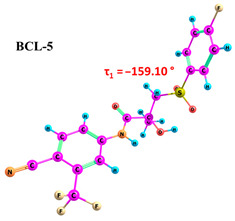
		O-H...OS	−31.05	2.740	1.895	144.11	
6	30.05	N-H...OH	−23.92	2.638	2.104	110.64	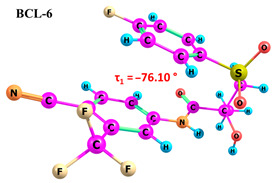
		N-H...OS	−9.04	3.095	2.450	120.92	
7	30.26	N-H...OH	−24.81	2.628	2.082	111.54	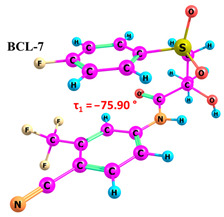
		N-H...OS	−9.03	3.072	2.463	118.09	
8	30.30	N-H...OS	−44.67	2.760	1.761	165.19	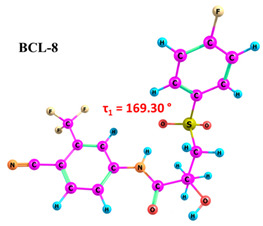
		O-H...OC	−42.00	2.547	1.881	132.18	
9	30.61	N-H...OH	−30.40	2.534	1.951	113.81	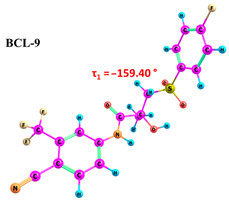
		O-H...OS	−31.22	2.739	1.893	144.30	
10	31.02	N-H...OH	−30.32	2.538	1.953	113.97	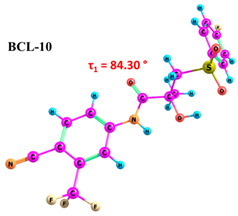
		O-H...OS	−30.60	2.751	1.896	145.69	

**Table 2 ijms-25-08254-t002:** Reference and conformation-determining distances were obtained using averaging models based on the conformer structures from quantum chemical calculations.

Conformer No	H12a-H12b, Å	H15/17-H14/18, Å	H12b-H14/18, Å	
BCL-1	1.78	2.81	4.33	
BCL-2	1.78	2.80	4.31	
BCL-3	1.79	2.81	3.44	
BCL-4	1.79	2.81	3.45	
BCL-5	1.78	2.81	3.62	
BCL-6	1.77	2.78	4.11	
BCL-7	1.77	2.79	4.11	
BCL-8	1.77	2.81	3.34	
BCL-9	1.78	2.81	3.61	
BCL-10	1.77	2.81	2.97	
Aver. Val.	1.78	2.80	Open	Close
			3.35	4.21

**Table 3 ijms-25-08254-t003:** Internuclear distances obtained from the structures of BCL conformers observed in various solid and bioactive forms.

Structure	H12a-H12b, Å	H14/18-H15/17, Å	H12b-H14/18, Å	
JAYCES01	1.56	2.60	3.53	
JAYCES02	1.56	2.59	4.25	
FAHFIG	1.56	2.59	3.16	
KIHZOR	1.56	2.61	4.33	
KIHZIL	1.57	2.60	3.43	
1Z95	1.85	2.76	4.03	
JAYCES	1.58	2.62	3.49	
Aver. Val.	1.59	2.62	Open	Close
			3.38	4.19

## Data Availability

Data is contained within the article and Appendix A.

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
