# Peer review of "Influence of Solvent Polarity on the Conformer Ratio of Bicalutamide in Saturated Solutions: Insights from NOESY NMR Analysis and Quantum-Chemical Calculations"

_ijms, 2024, doi:10.3390/ijms25158254_

Round 1

Reviewer 1 Report

Comments and Suggestions for Authors

The manuscript describes a combined theoretical (quantum-chemical calculations) and experimental (NMR spectroscopy) investigation of conformational preference of bicalutamide in solutions.
The compound is the known anti-cancer drug and the results of the investigation can potentially influence the respective field of research.
Regarding the text I have several comments and questions. I can recommend to accept the manuscript provided they are answered appropriately.

1. The Introduction is superfluously long with lots on information, details and references, which do not have a direct relation to the very specific problem treated in the work. I recommend to shorten this part.

2. Page 4, lines 168-189, this part contains essentially student-textbook-type information about idealized hybridizations of atoms and their corresponding geometrical arragements.
   This may be copletely omitted. Also I did not understand why "3-(trifluoromethyl)phenyl propamide" is written in line 168 and the reference [66] to flutamide is provided, whereas
   the authors refer to bicalutamide in Figure 1.

3. Table 1. How energies of hydrogen bonds E_HB have been calculated?

4. I principal question about NOESY analysis.
   If I understand correctly, results of quantum-chemical calculations have been directly used for interpretation of results.
   In particular, theoretical distances in several suitable pairs between hydrogen atoms were in utilized.
   The authors find substantial differences between data for solutions in CHCl3 and DMSO.
   On the other side, quantum-chemical calculations have been done without taking into account solvent effects.
   May this influence the results of the data analysis and lead to systematic errors?

5. Why not to perform quantum-chemical calculations for solutions in CHCl3 and DMSO and to check whether the
   predicted theoretical conformational preferences, for example based on Bolzmann distribution obtained from Gibbs energies,
   correspond to the experimental findings? Such a mini-benchmark of a selected theoretical approximation would significantly
   increase the importance of the paper.

6. Page 20, line 606. Why APFD functional has been chosen?

7. The Conclusions are again very lengthy and mostly duplicate the Introduction. A concise and concrete style would be much better for reader.

Author Response

We would like to thank the reviewer for their detailed and insightful comments on our manuscript. We appreciate the opportunity to address each of the points raised and provide clarifications and improvements to our manuscript. Below, we respond to the reviewer's comments point-by-point:

Comment 1: The Introduction is superfluously long with lots on information, details and references, which do not have a direct relation to the very specific problem treated in the work. I recommend to shorten this part.

Response 1: We acknowledge the reviewer's concern regarding the length of the Introduction. We have revised this section to focus more concisely on the specific problem addressed in our study, removing extraneous information and references that are not directly relevant. The revised Introduction now provides a more streamlined overview of the background and significance of our research.

Comment 2: This part contains essentially student-textbook-type information about idealized hybridizations of atoms and their corresponding geometrical arrangements. This may be completely omitted. Also, I did not understand why "3-(trifluoromethyl)phenyl propamide" is written in line 168, and the reference [66] to flutamide is provided, whereas the authors refer to bicalutamide in Figure 1.

Response 2: We have omitted the textbook-like information about hybridizations to keep the focus on our findings. Additionally, we corrected the inconsistency regarding "3-(trifluoromethyl)phenyl propamide" and ensured all references to bicalutamide are accurate and consistent throughout the manuscript.

Comment 3: How energies of hydrogen bonds E_HB have been calculated?

Response 3: We apologize for not providing sufficient details initially. We have now included a detailed explanation of the computational methods used to calculate these energies in the revised manuscript, ensuring clarity for the reader.

Comment 4: If I understand correctly, results of quantum-chemical calculations have been directly used for interpretation of results. In particular, theoretical distances in several suitable pairs between hydrogen atoms were utilized. The authors find substantial differences between data for solutions in CHCl3 and DMSO. On the other side, quantum-chemical calculations have been done without taking into account solvent effects. May this influence the results of the data analysis and lead to systematic errors?

Response 4: The reviewer correctly pointed out that our initial quantum-chemical calculations did not account for solvent effects, which could potentially influence our results. We have now performed additional quantum-chemical calculations incorporating solvent effects using the PCM model for both CHCl3 and DMSO. The revised analysis, including these new calculations, has been included in the manuscript. 

Comment 5: Why not perform quantum-chemical calculations for solutions in CHCl3 and DMSO and check whether the predicted theoretical conformational preferences, for example based on Bolzmann distribution obtained from Gibbs energies, correspond to the experimental findings? Such a mini-benchmark of a selected theoretical approximation would significantly increase the importance of the paper.

Response 5: We appreciate the reviewer's suggestion to perform quantum-chemical calculations for solutions in CHCl3 and DMSO. As noted in our response to point 4, we have now conducted these calculations and included a comparative analysis based on the Boltzmann distribution of Gibbs energies. This mini-benchmark significantly strengthens our findings.

Comment 6: Why has the APFD functional been chosen?

Response 6: The APFD functional was chosen for its accuracy in predicting non-covalent interactions, which are crucial for our study. We have included a citing studies that demonstrate its suitability for similar analyses.

Comment 7: The Conclusions are again very lengthy and mostly duplicate the Introduction. A concise and concrete style would be much better for the reader.

Response 7: We have revised the Conclusions to be more concise, focusing on the key findings and their implications. Redundant information has been removed to ensure a clear and succinct summary of our work.

Once again, we thank the reviewer for their valuable comments and suggestions, which have greatly improved the quality of our manuscript. We believe that the revisions made in response to these comments have strengthened our study and enhanced its clarity and impact.

Reviewer 2 Report

Comments and Suggestions for Authors

The manuscript presents an interesting experimental investigation into the conformational population of bicalutamide. The study also employs basic quantum-mechanical calculations to better analyze the results. While the findings are relevant to the scientific community, I recommend addressing a few issues to enhance clarity and reproducibility:

  1. On Pages 3 and 4, the authors refer to the C5-S8-C11-C12 dihedral angle. However, Figure 1 does not align with this numbering. Additionally, in Figure 9, distances involving H12b-H14/18, H15/17, and H14 are described as they appear in Tables 2 and 3. In addition, I encountered different numbering when examining Cartesian coordinates from the supplementary material. To improve consistency, I suggest adopting a single numbering scheme throughout the paper, ideally compatible with the supplementary information.
  2. For clarity, could the authors provide Cartesian coordinates for JAYCES01, JAYCES02, FAHFIG, KIHZOR, KIHZIL, 1Z95, and JAYCES structures in the supplementary material?

My primary concern is related to the fact that the quantum-mechanical calculations are performed in the gas phase, given that the structure’s conformation depends on solvent polarity. Moreover, the substantial energy difference between open and closed structures seems incompatible with experimentally measured ratios. To address this, I recommend computing geometries and energies in solution. This could shed light on the presence of the two conformers.

Author Response

We are grateful for the reviewer's thoughtful comments and recommendations on our manuscript. We have addressed each issue raised to enhance the clarity and reproducibility of our study. Our detailed responses are as follows:

Comment 1: On Pages 3 and 4, the authors refer to the C5-S8-C11-C12 dihedral angle. However, Figure 1 does not align with this numbering. Additionally, in Figure 9, distances involving H12b-H14/18, H15/17, and H14 are described as they appear in Tables 2 and 3. In addition, I encountered different numbering when examining Cartesian coordinates from the supplementary material. To improve consistency, I suggest adopting a single numbering scheme throughout the paper, ideally compatible with the supplementary information.

Response 1: We appreciate the reviewer's suggestion regarding the numbering scheme. We have thoroughly reviewed and revised the manuscript to adopt a consistent numbering scheme across all figures, tables, and text, aligning with the supplementary information. This change ensures uniformity and clarity throughout the paper.

Comment 2: For clarity, could the authors provide Cartesian coordinates for JAYCES01, JAYCES02, FAHFIG, KIHZOR, KIHZIL, 1Z95, and JAYCES structures in the supplementary material?

Response 2: We have included the Cartesian coordinates for the structures JAYCES01, JAYCES02, FAHFIG, KIHZOR, KIHZIL, 1Z95, and JAYCES in the supplementary material. This addition provides complete information for reproducibility and verification of our computational results.

Comment 3: My primary concern is related to the fact that the quantum-mechanical calculations are performed in the gas phase, given that the structure’s conformation depends on solvent polarity. Moreover, the substantial energy difference between open and closed structures seems incompatible with experimentally measured ratios. To address this, I recommend computing geometries and energies in solution. This could shed light on the presence of the two conformers.

Response 3: We agree with the reviewer's concern regarding the impact of solvent polarity on the conformational preferences of bicalutamide. We have now performed additional quantum-mechanical calculations considering solvent effects using the PCM model for relevant solvents. The energies have been recalculated in solution, and the new results are included in the revised manuscript. 

We thank the reviewer for their constructive feedback, which has significantly contributed to improving the clarity, consistency, and scientific robustness of our manuscript. We believe these revisions have enhanced the quality of our work and addressed the concerns raised.

Reviewer 3 Report

Comments and Suggestions for Authors

The manuscript entitled “Influence of Solvent Polarity on the Conformer Ratio of Bicalutamide in Saturated Solutions: Insights from NOESY NMR Analysis and Quantum-Chemical Calculations” submitted by V Sobornova and coworker presents a combined experimental and theoretical investigation about the conformational stability of the bicalutamide in solvent environments by using advanced NMR and ab initio molecular modeling techniques. The research topic explored in the manuscript looks of having a significant importance since it is a systematic and exhaustive investigation about the nature of the intramolecular bonding and conformation inside the crystal structure. The manuscript is well written and contains an interesting methodological discussion and analysis. Both experimental and molecular modelling techniques have been applied at the highest scientific level, and the results obtained with these methods are of great interest

General observation:

-          How the E_HB intramolecular hydrogen bond energies shown in Table 1 were calculated? A short description about the method should be included in the manuscript.

-          Was the solvent model included in their theoretical calculations? It is not mentioned in the “Materials and Methods” section. If Yes, in Table 1 what solvent case is shown?

The manuscript is suitable for publication in the IJMS journal, but it needs for a short major revision.

Author Response

We would like to express our gratitude to the reviewer for their positive evaluation and constructive comments on our manuscript . We are pleased to address the reviewer's comments and provide the necessary revisions to enhance the clarity and completeness of our study.

Comment 1: How the E_HB intramolecular hydrogen bond energies shown in Table 1 were calculated? A short description about the method should be included in the manuscript.

Response 1: We appreciate the reviewer's suggestion to provide more details on the calculation of E_HB intramolecular hydrogen bond energies. We have now included a short description of the computational method used to calculate these energies in the “Materials and Methods” section. 

Comment 2: Was the solvent model included in their theoretical calculations? It is not mentioned in the “Materials and Methods” section. If Yes, in Table 1 what solvent case is shown?

Response 2: The reviewer's observation is correct, and we apologize for the oversight. We have now clarified in the “Materials and Methods” section that a Polarizable Continuum Model (PCM) was used to account for solvent effects in our quantum-chemical calculations. The solvents for which the calculations were performed (e.g., CHCl3 and DMSO) are now clearly indicated, providing complete information to the reader.

We believe that these revisions address the reviewer's concerns and enhance the clarity and scientific rigor of our manuscript. We appreciate the reviewer's positive assessment of our work and their constructive feedback, which has been invaluable in refining our study.

Round 2

Reviewer 1 Report

Comments and Suggestions for Authors

Thank you for your answers and corrections.

Comments on the Quality of English Language

The paper is understandable but some minor points related rather to style may be polished. This is, however, a non-blocking issue and the paper may be published as is.

Author Response

Comment #1: The paper is understandable but some minor points related rather to style may be polished. This is, however, a non-blocking issue and the paper may be published as is.

Response #1: Thank you for your feedback. We're pleased to hear that the paper is understandable. We've addressed some of the stylistic issues you noted and made the necessary adjustments. We appreciate your suggestion and support for the paper's publication.

Reviewer 2 Report

Comments and Suggestions for Authors

The manuscript has significantly improved, and I recommend its publication with two minor suggestions. Firstly, in Figure 3a, setting a shorter range for the y-axis would help to highlight the differences in energy more clearly. At present, all the energies appear identical, forming a nearly horizontal line. Secondly, for equation 1, I believe the second half should have a proportional sign rather than an equal sign. Could the authors please double-check this equation?

Author Response

Thank you for your positive feedback and valuable suggestions. We're pleased to hear that you find the manuscript significantly improved. Regarding your suggestions:
Comment #1: Firstly, in Figure 3a, setting a shorter range for the y-axis would help to highlight the differences in energy more clearly. At present, all the energies appear identical, forming a nearly horizontal line. 
Response #1: We agree that adjusting the y-axis range could enhance the clarity of the energy differences. We will revise the figure to set a shorter range on the y-axis, which should better highlight the distinctions between the energy values.
Comment #2: Secondly, for equation 1, I believe the second half should have a proportional sign rather than an equal sign. Could the authors please double-check this equation?
Response #2: We appreciate your careful review of the equation. We replaced the equal sign with a proportional sign, ensuring the equation accurately reflects the intended relationship.
Thank you again for your insightful comments.

Reviewer 3 Report

Comments and Suggestions for Authors

This new revised version of the manuscript shows significant improvements compared to the first submitted version of the manuscript. The authors have given satisfactory solutions and explanations for the addressed questions related to their original work.

Author Response

Thank you for your positive feedback. We appreciate your recognition of the improvements made in the revised manuscript. Your comments and questions greatly contributed to enhancing the clarity and quality of our work. We are glad that our responses were satisfactory and addressed the concerns raised.